# Effect of Carbon Nanofibers on the Strain Rate and Interlaminar Shear Strength of Carbon/Epoxy Composites

**DOI:** 10.3390/ma16124332

**Published:** 2023-06-12

**Authors:** Paulo Santos, Abílio P. Silva, Paulo N. B. Reis

**Affiliations:** 1C-MAST—Centre for Mechanical and Aerospace Science and Technologies, University of Beira Interior, 6201-001 Covilhã, Portugal; abilio@ubi.pt; 2University of Coimbra, CEMMPRE, ARISE, Department of Mechanical Engineering, 3030-780 Coimbra, Portugal

**Keywords:** carbon fibers reinforced polymers (CFRP), carbon nanofibers (CNFs), corrosive solutions, interlaminar shear strength, strain rate, static properties

## Abstract

The static bending properties, different strain rates and interlaminar shear strength (ILSS) of carbon-fiber-reinforced polymers (CFRP) with two epoxy resins nano-enhanced with carbon nanofibers (CNFs) are studied. The effect on ILSS behavior from aggressive environments, such as hydrochloric acid (HCl), sodium hydroxide (NaOH), water and temperature, are also analyzed. The laminates with Sicomin resin and 0.75 wt.% CNFs and with Ebalta resin with 0.5 wt.% CNFs show significant improvements in terms of bending stress and bending stiffness, up to 10%. The values of ILLS increase for higher values of strain rate, and in both resins, the nano-enhanced laminates with CNFs show better results to strain-rate sensitivity. A linear relationship between the logarithm of the strain rate was determined to predict the bending stress, bending stiffness, bending strain and ILSS for all laminates. The aggressive solutions significantly affect the ILSS, and their effects are strongly dependent on the concentration. Nevertheless, the alkaline solution promotes higher decreases in ILSS and the addition of CNFs is not beneficial. Regardless of the immersion in water or exposure to high temperatures a decrease in ILSS is observed, but, in this case, CNF content reduces the degradation of the laminates.

## 1. Introduction

Carbon-fiber-reinforced polymers (CFRP) are characterized by their low weight, high strength and stiffness and low coefficient of thermal expansion in the fiber direction, making them attractive for a wide range of applications including wind energy generation structures, transportation, aerospace and sports equipment. Superior properties, excellent strength-to-weight ratio and versatility in manufacture/application, particularly in the transport sector, result in reduced energy consumption, which is the main driver for the use of CFRP [1,2].

Several methods are used in the manufacture of CFRP for different applications, depending on the final shape, size and geometry of the component or structure. Some of the main methods used to manufacture composites are hand lay-up, vacuum bagging, vacuum-assisted resin transfer molding (VARTM), vacuum-assisted resin infusion molding (VARIM), autoclave, compression molding, pultrusion and filament winding [1,2,3].

Carbon fibers are an important constituent of CFRP and have played a central role in the production of lightweight high-performance composites due to their inherent properties—high tensile strength, high modulus, low densities, good thermal and electrical conductivity, high thermal and chemical stabilities, low coefficient of thermal expansion, biological compatibility and fatigue resistance—which are imparted into the properties of the final composite [4,5,6].

As far as the matrix is concerned, the first choice in engineering is epoxy resin, which can be characterized as a molecule containing more than one epoxy group capable of being converted into a thermoset form. It has high electrical resistivity and good performance at elevated temperatures, thanks to both its higher heat deflection temperature and high glass transition temperature (*T_g_*). It also exhibits optimum adhesion to various substrates and to fibers used as reinforcements in composite materials [7,8]. In terms of application, it is very versatile and, for this, epoxy resins are among the mainstream plastic materials for applications such as coatings, adhesives, laminates and structural components. However, when the application is structural, they may prove to be either brittle, notch-sensitive or both. As a result, considerable effort has been devoted to improving the toughness of epoxy resins, particularly in the last two decades [9]. One of their characteristics, especially in the case of low-viscosity liquids, is that their properties can be optimized by mechanical means, as they can be nano-enhanced with nanoparticles.

The reinforcing capacity of nanofibers increases with decreasing their dimensions, i.e., nanofibers have higher reinforcing capacity than microfibers and short fibers due to the increased interfacial area [10,11]. This has the ability to strengthen the fiber/polymer interface and increase matrix stability [12], thereby improving their mechanical, chemical and/or physical properties. Nowadays, the strongest materials available for structural applications are in the form of small particles: nanoparticles (nano-powder, nanoclusters, nanocrystals) are sized less than 100 nm in diameter; nanofibers have diameters less than 100 nm and a high aspect ratio; and nanoplatelets are 2D stacks of nanomaterials that may be made from metals, ceramics or graphene (GP).

In addition to being the most expensive, GP is actually the most commonly used in composites manufacturing, for example; it has 1 to 10 nm thick and 1 to 15 μm in diameter, resulting in aspect ratios of up to 1:1000 and surface areas up to 700 m^2^/g. Nanocarbon comprises many allotropes; these are the same element, carbon, but with different arrangements of atoms, and include carbon and graphene dots, graphene sheets and platelets, and graphene rolled into high-aspect-ratio carbon nanotubes (CNTs), sphere-like buckyballs/fullerenes and multi-layered nano-onions [13].

On the other hand, carbon nanofibers (CNFs), vapor-grown carbon fibers (VGCFs) or vapor-grown carbon nanofibers (VGCNFs) are cylindrical nanostructures (1D) with graphene layers arranged as stacked cones, cups or plates [14,15,16]. They contain more reactive carbon edges that can interact with the matrix better, and they are an alternative for manufacturing single-filled, bi-filled and multi-filled synergistic reinforced composite structures for shielding purposes [17]. However, nowadays, CNFs of different morphologies have been synthesized, such as the following: platelet-like CNFs composed of small graphene layers perpendicular to the fiber axis; fishbone-like CNFs or herringbone CNFs where the graphene layers are inclined with respect to the fibril axis; CNFs ribbons are comprised of straight, unrolled graphene layers that are parallel to the fibril axis with noncylindrical cross-sections; stacked-cup CNFs which are a continuous layer of rolled (spiral) graphene along the fiber axis; and thickened tubular CNFs comprising a base structure of one of the previously mentioned catalytic nanofilaments (CNFs or CNTs) with a variable coating of amorphous carbon [18].

CNFs containing carbon with diameters of 50 to 200 nm, with exceptional mechanical and electrical properties, can be prepared mainly by two methods: one is catalytic thermal chemical vapor deposition growth, and the other is electrospinning followed by heat treatment [15,19]; relatively large quantities use a variety of metals either in powder or supported form as catalytic entities. The efficiency of load transfer between nanomaterial and polymer chains is a critical factor for the mechanical properties and consequently for the structural behavior of the composite. If this transfer is complete and the percentage of nanomaterials dispersed in the polymer is optimized, the behavior of the composite will be improved due to the increase in its mechanical properties. This enhancement is related to some nano effects of the nanomaterials, such as their high aspect ratio and large interfacial area, which can form a huge interphase between nanomaterials and the matrix and, thus, contribute positively to the load transfer [20].

Numerous studies combine these materials to optimize them, whether for increasingly demanding applications or to replace those already applied. For example, in the investigation by Zhou et al. [21], CNFs were infused in epoxy with a high-intensity ultrasonic irradiation method. The adding of 2.0 wt.% CNFs and using VARTM to produce carbon-fiber-reinforced epoxy composites led to increases of 17.4% in tensile strength and 22.3% in flexural strength compared to the control composite. Green et al. [22], applying the VARIM technique, produced laminates of E-glass fibers reinforced with 0.1 wt.% and 1 wt.% of the multiscale fiber-reinforced composites (M-FRCs) based on CNFs dispersed in an epoxy resin. The tests showed that the flexural strength increased by 16 and 20%, the modulus increased by 23 and 26% and the ILSS properties increased by 6% and 25%, respectively, for the 0.1 wt.% and 1 wt.% M-FRCs when compared to the control FRCs. Arai et al. [23] produced two types of unidirectional CFRP laminates with interlayers made by CNFs with a volume fraction of about 10%, inserted between the carbon prepregs with sixteen or thirty-two layers using the autoclave technique. The interlaminar strength of unidirectional composites was studied and, by inserting a CNFs interlayer, the critical curve of the damage criterion increased by about 20% in the area where the compression stress acts in the transverse direction of the beam. In the work by Palmeri et al. [24], 0.69 and 2.0 wt.% CNFs were dispersed in the epoxy matrix phase of carbon-fiber-reinforced composites in unidirectional and quasi-isotropic configurations. Tensile modulus and strength in the quasi-isotropic composites containing CNFs were enhanced without reducing strain to failure, and short-beam shear strength (SBSS) was increased by 15 to 22% with the addition of CNFs in both composite orientations. Bortz et al. [14] dispersed 1.0 wt.% CNFs in the epoxy matrix first by hand-mixing, and afterwards it was passed through a three-roll calender mill to manufacture a continuous carbon-fiber-reinforced composite via the VARTM technique. Improvements were observed in the flexural modulus and flexural strength. Tensile failure strain was also found to increase from 10.7% to 11.9%, indicating simultaneous stiffness, strength and toughness enhancements. Zhou et al. [25] used the VARTM technique to fabricate unidirectional carbon/epoxy composites modified with 2.0 wt.% CNF composites. Composites prepared with CNFs displayed a 15.8% increase in ILSS in comparison to control composites. Ma et al. [26] applied the filtering-membrane-assisted method to obtain composites of carbon/epoxy with high content of CNFs and improve the interlaminar properties. The ILSS of CFRP made by this method increased by up to 55% at 3.0 wt.% CNFs, while the maximum of ILSS increases up to 11% at 1.0 wt.% CNFs by the no-mem-brane-assisted method. The results of Dhakate et al. [16] indicated that the inclusion of 1.1 wt.% CNFs at the interlaminar region between carbon fiber fabric layers in the epoxy matrix of the carbon laminate improves the mechanical properties compared to control laminate and for higher values of CNFs, the properties are affected. The bending strength increased by 175%, modulus by 200%, and ILSS by 190%. Recently, Senthil et al. [27] in your work CNFs were filled into the glass fiber/unsaturated polyester composite, by a combined hand lay-up technique followed by a filtering membrane vacuum-assisted method. For a concentration of 2 wt.% CNFs reinforcement shows an 89% increase in the ILSS of the composite. The composite with 2 wt.% of CNFs also exhibits enhanced storage elastic modulus and hardness to about 49% and 26%, respectively. In the work by Ramezani et al. [28], cross-ply laminated composite specimens were fabricated with the stacking sequence of [0_2_/90_6_]_s_ using unidirectional E-glass fibers and the effect of adding CNFs on the reduction of matrix cracking was studied. The specimens were tested under tensile loading, and it was concluded that the addition of CNFs fillers into the composite specimens resulted in lower crack densities and less stiffness reduction at certain applied stress levels by adding 0.25 wt.% CNFs into the matrix.

However, due to the rapid growth in the use of composite materials, they can be exposed to a range of corrosive environments and temperature variations during their service life, causing degradation of material properties and affecting their static stability and long-term durability [29,30,31]. Several immersion aging studies of CFRP composites immersed into water and acidic or alkaline solutions at different temperatures show that degradation adversely affects the mechanical properties and, according to thermal and mechanical analyses, ageing depends on the ageing temperature and the ageing medium, being more pronounced at higher temperatures, mainly in acidic conditions [29,30]. Uthaman et al. [29] attributed the decreases in the properties of the composites to the degradation of the resin matrix and debonding at the fiber–resin interface. On the other hand, Yang et al. [30] showed that the addition of multi-walled carbon nanotubes (MWCNTs) improves the ageing resistance of CFRPs due to good interfacial interaction and their high barrier property. Nanoclays, in turn, improve the ageing resistance of CFRPs due to their high aspect ratio and moderate interfacial adhesion. In short, CFRPs containing nanofillers reduce the loss of mechanical properties less than pure CFRP. Kojnoková et al. [31] studied how different chemical environments at a given temperature affect the viscoelastic properties of composites when subjected to degradation by immersion and confirmed a synergistic influence caused by degradation changes and a plasticizing effect due to water absorption, which causes a reduction in the modulus of elasticity.

Sinmazçelik et al. [32] studied the ILSS changes in unidirectional carbon-fiber-reinforced polyetherimide (PEI) composites following exposure to various liquid environments, i.e., 0.6 molar NaCl, triple-distilled water (TDW) and petrol, and a temperature of 90 °C for 90 days, highlighting a huge decrease in the results for all samples subjected to liquid environments. For example, the ILSS of control samples was 85.28 MPa, and the results of samples immersed in NaCl environments after 90 days were 63.18 MPa, which represents a decrease of 39.2%. Mahato et al. [33] engaged in various experimental analyses of the effect of thermal shock on ILSS of thermally (above and below the room temperature) conditioned woven-fabric glass/epoxy and glass/polyester composites, concluding that there is a decrease in ILSS value with an increase in conditioning time. Kopietz et al. [34] studied the impact of different media—water, sodium hydroxide (NaOH) and hydrochloric acid (HCl)—on ILSS of glass-fiber-reinforced plastics (GFRP), and a negative impact from all media—water, acid and alkaline solutions—was clearly verified on all tested composites.

Zhang et al. [35] studied the influence of environmental factors such as water, seawater, acid, alkali and organic solutions under post-cure and pre-cure curing conditions on the ILSS of cured structural carbon-fiber-reinforced epoxy composite. The results showed a reduction of the ILSS properties, and the control of various contaminants such as water, acids, alkalis, salts and organic solvents can have significant effects on the mechanical performance of laminate composite components during the manufacturing process and their usage. Silva et al. [36] investigated how accelerated ageing under different conditions (distilled water, seawater, UV radiation plus water spray) for up to 3000 h affected SBS of pultruded CFRP rods with epoxy and vinylester matrices. The experimental results showed that resistance SBS was strongly affected by seawater and distilled water ageing. SEM images showed greater fiber/matrix adhesion for the carbon/epoxy composites, extensive degradation and microcracking of the vinylester matrix and debonding of the fiber/matrix interface in the carbon/vinylester rods.

Nowadays, both industry and researchers are faced with new challenges/opportunities on a daily basis. Due to the currently changing climate, the paradigm shift in production/distribution/energy consumption, changes in the way we travel and the need for more protection, both individual and collective, to prevent natural or provoked attacks, the application/development/optimization of composite materials is topical and requires continuous advancement to respond to all these demands. In the future, advanced materials will be responsible for achieving climate neutrality, thereby boosting the economy through green technologies, the development of sustainable transport and safety in all areas of human endeavor. In this sense, optimized nanocomposite materials will be a very important part of the answer to problems in industrial applications, providing materials that are easier to manufacture in complex shapes, lighter, structurally stronger, corrosion-resistant regardless of the environment, with low thermal conductivity and a longer life cycle, reducing their environmental footprint and recycling issues.

Therefore, in this work, an initial characterization of the static properties in the bending mode of the laminates manufactured with two different epoxy resins will be carried out, and the influence of the strain rate on the mechanical properties is studied, which will allow us to understand the influence of different wt.% CNFs on the composites in response to static demands. The ILSS test will allow us to understand the benefits of adding CNFs and the response of the laminates in terms of adhesion between fiber and matrix. The effect of water temperature, concentration and temperature of hydrochloric acid (HCl) and sodium hydroxide (NaOH) solutions on ILSS strength after immersion will be analyzed.

This study aims to contribute to improving the scientific knowledge of the effect of adding CNFs as a low-cost reinforcement through the application of simple dispersion techniques in composites. It responds to a gap in the literature that does not report studies on the effect of corrosive environments on ILSS strength. Although some studies have investigated the ageing behavior of CFRP with nanofillers in different solutions, the study of common ageing factors in the industry, such as hydrochloric acid (HCl), sodium hydroxide (NaOH), water and temperature are not frequent. The degradation of the mechanical properties of CFRPs under these different ageing conditions was systematically studied in order to understand their behavior and mathematical models are proposed to estimate the properties for complementary conditions.

## 2. Materials and Methods

### 2.1. Production of Composite Laminates

Eight plain weaves of carbon fibers at 0 and 90 degrees with 98 ± 4% g/m^2^ in each direction were combined with two different resins (epoxy resin SR 8100 with hardener SD 8824, both supplied by Sicom, Paris, France, and an epoxy resin AH 150 with hardener IP 430, both supplied by Ebalta, Rothenburg, Germany) to produce laminates with overall dimensions of 330 × 330 × *t* [mm] by hand lay-up technique. After curing, the thickness (*t*) obtained for each laminate was 1.5 ± 0.1 mm for laminates with Sicomin resin and 1.9 ± 0.1 for the Ebalta resin. The main mechanical and physical properties of the epoxy resins are summarized in Table 1, and more details about them can be found in [37]. The systems were placed inside a vacuum bag and subjected to a load of 2.5 kN to obtain a constant fiber volume fraction and uniform laminate thickness. Finally, composites involving the Sicomin matrix were cured at room temperature for 24 h and subjected to a post-cure at 40 °C for 24 h, while those involving the Ebalta matrix were cured at room temperature for 48 h and subjected to a post-cure at 80 °C for 5 h. During the first 4 h of the curing process, all composites were subjected to vacuum in order to eliminate any air bubbles.

The literature reports that the mechanical properties of commercial resins can be improved by using nano-reinforcements, but this is strongly dependent on the nano-particle content and type of resin [38]. In this context, to evaluate the benefits achieved in terms of bending properties and interlaminar shear strength (ILSS), similar composite laminates were produced with different weight contents of CNFs. These values were 0.25, 0.5, 0.75 and 1 wt.%, which are similar to those used by the authors in another study that aimed to maximize the static properties of these nano-reinforced resins [39].

Therefore, in terms of bending response, this study aims to compare the CNF content that maximized these properties in the commercial resins with that obtained for the composite laminates. In both studies, the CNFs used were supplied by Sigma-Aldrich (Burlington, MA, USA) and are pyrolytically stripped (conical) with an average diameter of 130 nm, a length between 20 and 200 μm and an average specific surface area of 54 m^2^/g. In terms of manufacturing process, resin and CNFs were initially mixed using a high-speed shear mixer at 1000 rpm at room temperature for 3 h, followed by 10 min at 150 rpm for the hardener to disperse into the system. To improve the CNF dispersion, this process was combined with an ultrasonic bath (with a frequency of 40 kHz). Finally, the system was degassed in a vacuum oven. All this methodology is detailed in [40]. Subsequently, the nano-reinforced resins were combined with the carbon fiber fabrics using the methodology described above and summarized in Figure 1.

### 2.2. Experimental Tests

From those laminated plates, two batches of samples were obtained with dimensions of 80 × 10 × *t* (mm) and 12 × 4 × *t* (mm) for the bending and interlaminar shear tests, respectively. The static three-point bending (3PB) tests were carried out at room temperature on a Shimadzu universal testing machine, model Autograph AGS-X (Shimadzu, Kyoto, Japan), equipped with a 10 kN load cell. For each condition, six specimens were tested in accordance with the European Standard EN ISO 178:2003 and at a displacement rate of 2 mm/min. Finally, the bending properties (stress, stiffness and strain) were obtained using the following equations:(1)$$\sigma =\frac{3\;P\;L}{2\;b\;h^{2}}$$
(2)$$E=\frac{\Delta P\;L^{3}}{48\;\Delta u\;I}$$
(3)$$\varepsilon_{f}=\frac{6\;S\;h}{L^{2}}$$
where *P* is the load, *L* is the span length, *b* is the width, *h* is the thickness of the specimen, *I* is the moment of inertia of the cross-section, *ΔP* and *Δu* are, respectively, the load range and flexural displacement range in the middle span for an interval in the linear region of the load versus displacement plot and *S* is the deflexion. The bending modulus was obtained by linear regression of the load–displacement curves considering the interval in the linear segment with a correlation factor higher than 95% [41].

The strain rate effect was analyzed only with laminates whose CNF contents maximized the bending properties and, for this purpose, displacement rates of 200, 20, 2, 0.2 and 0.02 mm/min were used, which correspond to strain rates ($\dot{\varepsilon }$) of 1.2 × 10^0^, 1.2 × 10^−1^, 1.2 × 10^−2^, 1.2 × 10^−3^ and 1.2 × 10^−4^ s^−1^ for laminates with a Sicomin matrix and 1.4 × 10^−1^, 1.4 × 10^−2^, 1.4 × 10^−3^, 1.4 × 10^−4^ and 1.4 × 10^−5^ s^−1^ for laminates with an Ebalta matrix, values that were obtained by the Equation (4):(4)$$\dot{\varepsilon }=\frac{d\varepsilon_{f}}{dt}=\frac{6\;V_{T}\;b}{L^{2}}$$
where $\dot{\varepsilon }$ is the peripheral fiber strain, *t* is the time, *V_T_* is the crosshead speed, *L* is the span length and *h* is the thickness of the specimen.

In terms of interlaminar shear strength (ILSS), the short-beam shear (SBS) method is the simplest and most widely used. Therefore, interlaminar shear tests were carried out according to ASTM D2344/D2344M-00 standard using the same equipment (Shimadzu Autograph AGS-X) with a crosshead speed of 1 mm/min. The ILSS value is obtained by Equation (5):(5)$$\tau_{S}=\frac{3}{4}\frac{P_{C}}{w\;h}$$
where $P_{C}$ is the maximum load and *w* and *h* are, respectively, the width and the thickness of the beam. For each condition, five samples were tested at room temperature using the same equipment of the 3PB static tests. The nominal dimensions of the specimen for ILSS tests are a length of 12 mm, width (*w*) and height (*h*) approximately 4 mm and 2 mm, respectively, and a distance between the supports in the specimens of 10 mm, all a function of the height *h* of the laminate. The loading rate effect on the interlaminar shear strength (ILSS) of the laminates was also studied for displacement rates of 0.01, 0.1, 1, 10, 100 and 1000 mm/min.

Figure 2 shows the apparatus, schematic view of the tests and respective dimensions of the samples used in the experimental tests.

From the batch of samples that were produced for use in the interlaminar shear tests, a part of them was tested as produced, while the other set was immersed into different hostile solutions immediately after production. The hostile solutions selected were hydrochloric acid (HCl) and sodium hydroxide (NaOH) because they aim to simulate the different environments that can be found in the civil engineering sector or in the chemical and food industry [42] in addition to being among those that most affect the mechanical properties of composite materials [43,44,45] Therefore, some samples were completely submerged into hydrochloric acid (HCl) and sodium hydroxide (NaOH) at room temperature for 20 days. Considering these conditions, concentrations of 5%, 15%, 25% and 35% by weight (wt.%) were analyzed for both solutions in order to evaluate their effect on the interlaminar shear strength (ILSS). Finally, the effect of temperature was also analyzed and, for this study, the specimens were previously submerged for 20 days at room temperature, 40 °C and 80 °C into both solutions at a concentration of 15 wt.%. For comparison purposes, some samples were also immersed into distilled water at room temperature, 40 °C and 80 °C. Finally, before testing, the specimens immersed into the different hostile solutions were washed with clean water and dried at room temperature.

Table 2 summarizes all the tests performed and conditions analyzed in this study, while Figure 3 schematically shows the methodology adopted to evaluate the effect of different hostile solutions on interlaminar shear strength.

## 3. Results and Discussion

For all composite laminates produced, 3PB static tests were carried out to evaluate the effect of CNF content on bending properties. Typical bending stress–strain curves are shown in Figure 4, which are representative of all others obtained for the same conditions.

These curves are characterized by a linear increase in bending stress with bending strain (linear elastic region) close to the maximum bending load, followed by an abrupt decrease due to the imminent collapse of the laminates. Figure 5 shows the typical damage mechanism observed in all composites, where broken fibers under compression are notorious, accompanied by small delaminations around them. The zigzag aspect observed in some curves is due to progressive breakage of various fibers, while the others show an abrupt collapse due to instantaneous breakage of a significant number of fibers under compression. In fact, it has been reported in the literature that this failure mode is typical of composites containing carbon fibers [46,47] due to the low compressive strength of these fibers. In addition, the high stress concentration that occurs in the contact region with the load pin leads to higher local compressive stresses and, consequently, higher propensity for an imminent collapse [46].

Therefore, the main bending properties were obtained from the curves shown in Figure 4 and summarized in Figure 6 for both resins. Regarding laminates produced with Sicomin resin (Figure 6a), the maximum bending strength (905.3 MPa) is reached for 0.75 wt.% of CNFs, which is 20.4% higher than that obtained for the control samples (752.2 MPa). When the filler content increases up to 1 wt.%, the bending strength decreases to 874.8 MPa. Similar behavior occurs for laminates produced with Ebalta resin but, in this case, the maximum bending strength (850.9 MPa) is reached for 0.5 wt.% of CNFs and is 12.5% higher than the value obtained for the control samples (756.2 MPa). Concerning the bending modulus, an increase in stiffness (13.8%) is observed with the increase in filler content for laminates with Sicomin resin (from neat resin to 1 wt.% of CNFs), while for laminates with Ebalta resin the maximum bending modulus is reached at 0.5% and is 8.8% higher than the value obtained for the control samples (47.5 GPa). However, when comparing the bending modulus for the maximum bending strength of both laminates, the value obtained for the one produced with Sicomin resin is 18.8% higher than the one produced with Ebalta resin. Finally, as expected, the bending strain decreases with increasing filler content for both resins.

These results are in good agreement with those obtained by Santos et al. [37], for whom the same values for the respective resins enhanced with CNFs were found. In this case, the maximum bending strength was also achieved for the same weight contents shown in Figure 6, evidence that maximizing the bending properties of the resins also maximizes the bending properties of the laminates produced with them. This denotes that, in composite materials, the matrix is the weakest phase and any increase in its mechanical properties increases those of the composite. On the other hand, if the use of nanoparticles is an effective strategy to increase strength and stiffness without compromising density, toughness and the manufacturing process [48], it is also observed that the improvements achieved are strongly related to the nano-reinforcement content and resin used [38].

According to Santos et al. [37], due to the higher viscosity of the Sicomin resin, it would be expected that the bending properties would be maximized for lower CNF contents than those used for the Ebalta resin. This is because the low viscosity of a resin allows better organization of nanoparticles [49]. However, based on the greater physicochemical compatibility of the Sicomin resin (greater acceptance of the filler by the matrix), they justified the values observed for the respective CNF contents. Finally, as shown in Figure 7, higher filler contents promote agglomerations/aggregations, which correspond to defects and act as stress-concentration points in nanocomposites [46,47,50].

In addition, the interfacial area between the polymer matrix and the nanoparticles also decreases and only a few polymer molecules can penetrate between the nanoparticles, promoting, in this case, a substantial increase in viscosity [51,52]. This evidence was well-reported in a previous study [39],where SEM images of the fracture surfaces revealed good dispersion for 0.5 wt.% CNFs, while for 0.75 wt.% agglomeration/aggregation occurs. Under these conditions, the mechanical properties are affected, especially in terms of bending strength.

Strain rate is the change in strain (deformation) with respect to time, and for this study the bending properties are analyzed only for the conditions that promoted the highest bending strength, i.e., laminates with Sicomin resin reinforced with 0.75 wt.% of CNFs and Ebalta resin reinforced with 0.5 wt.% of CNFs. Therefore, Figure 8 shows representative stress–strain curves for all strain rates obtained with laminates produced with Sicomin resin; however, they also reproduce the behavior obtained with the Ebalta resin.

It is noticed that the profile of these curves is very similar to that observed in Figure 4, evidencing similar damage mechanisms. The bending properties were also obtained, which are summarized in Figure 9. Symbols represent the average values, and the dispersion bands are the maximum and minimum values obtained from the experimental tests.

It is possible to observe that the maximum bending stress increases for all laminates with an increase in strain rate, which is in line with the literature [49,53]. For example, considering the laminates produced with Sicomin resin and the range of strain rate studied, it is found that the control laminate increases its bending stress by around 22.7%, while that produced with resin reinforced with 0.75 wt.% of CNFs is 15.4%. On the other hand, the same comparison for Ebalta resin leads to values of 21.6% and 25.8%, respectively. In terms of bending modulus, these values are 7.4% and 6.3% for Sicomin resin, and 3.9% and 18.4% for Ebalta resin. From this analysis, and regardless of resin, it is possible to observe that the control laminates have a very similar sensitivity (22.7% and 21.6%), but when the resins are nano-reinforced, while the laminates produced with Sicomin resin present lower sensitivity to the strain rate (from 22.7% to 15.4%), those produced with Ebalta resin present higher sensitivity (from 21.6% to 25.8%). According to the literature, these increases are related to secondary molecular processes, because an increasing strain rate decreases the molecular mobility of the polymer chains, making the chains stiffer [54].

Finally, as suggested by the literature [55,56,57], a linear model can be fitted to the data according to Equations (6)–(8), with which it is possible to obtain the sensitivity to strain rate through the slope of the curves [58].
(6)$$\sigma =a+b\times \dot{e}$$
(7)$$E=a+b\times \dot{e}$$
(8)$$\varepsilon =a+b\times \dot{e}$$

In these equations, σ is the maximum bending stress, E is the bending modulus, ε is the strain at maximum bending stress and $\dot{e}$ is the logarithm of strain rate and *a* and *b* constants presented in Table 1. Therefore, from Table 3, it is possible to observe that the proposed linear relationships (between the logarithm of strain rate and mechanical properties) present good precision (standard deviation (Stdev) higher than 0.951) and can be used as prediction tools.

The interlaminar shear strength test provides the composite’s resistance to delamination under shear forces parallel to the laminate layers. In many cases, the low interlaminar shear strength of polymer composites can be explained by the use of fibers without surface treatment or with inadequate treatments for the matrix used [59]. Therefore, to evaluate the interlaminar shear strength (ILSS) of the laminates produced with the different resins, as well as the benefits obtained with CNFs, this study used the short-beam shear test for this purpose. In this context, Figure 10 shows the load–displacement curves obtained for all configurations studied.

All curves obtained from the short-beam shear tests show a quasi-linear increase in the load with displacement, after which a plateau region appears instead of an abrupt drop in load. However, according to Espadas-Escalante and Isaksson [60], delaminations are more related to curves that show abrupt load drop, while the observed curve shape (containing plateaus) suggests a multi-step failure mode. In this case, delamination zones appear when two longitudinal yarns are in contact or when a longitudinal yarn and a transverse yarn are in contact. On the other hand, zones where two transverse yarns are in contact promote transverse yarn cracking, causing an apparent delamination in interlaminar shear strength. However, the literature also reports that interlaminar shear strength is very similar for both types of failure (different curve shapes) [61]. In this context, Figure 11 shows the damage mechanisms observed for laminates with neat Ebalta resin, which are also representative of all other laminates.

It is possible to observe delaminations resulting from the contact between two longitudinal yarns (A), another delamination resulting from the contact between a longitudinal yarn and a transverse yarn (B) and transverse yarn cracking (C) in a region where two transverse yarns are in contact. In the last case (C), the crack propagates further into a resin-rich region. These failure mechanisms created a region of coalescence of cracks in which one of them became dominant (A). Therefore, the yarns’ interactions and the resin-rich regions (favorable to cracking) prove to be determinants in triggering the damage and confirm what is referred to in the literature [61].

Based on these curves (load–displacement shown in Figure 10), He and Gao defined the total fracture work (*W_t_*), as the area under the curve, and verified its dependence with nanoparticle content [62]. A similar effect can be observed in this study, where *W_t_* depends on CNF content and type of resin. In terms of interlaminar shear strength, its value is also obtained from these curves and Figure 12 summarizes the effect of CNFs on this parameter. Symbols represent the mean values, and the scatter bands represent the maximum and minimum values. It is possible to observe that, in terms of laminates produced with neat resins (control laminates), the highest ILSS value is obtained with the Sicomin resin, 3.3% higher than that obtained with the Ebalta resin, evidencing its greater physicochemical compatibility in relation to carbon fibers used (as mentioned above). On the other hand, regarding the CNF content, its increase is responsible for higher ILSS values until reaching a maximum that depends on the resin and, consequently, on the CNF content. For example, while the maximum ILSS value is reached with 0.75 wt.% of CNFs in laminates produced with the Sicomin resin, in the case of those involving the Ebalta resin, it is 0.5 wt.%. Moreover, compared to the control laminates, the ILSS value for those using the Sicomin resin reinforced with 0.75 wt.% of CNFs is about 8.6% higher, while for the Ebalta resin reinforced with 0.5 wt.% it is about 9.4% higher.

These results are similar to those obtained in the static characterization, where the bending strength was also maximized for the same CNF content. Therefore, it can be concluded that these CNF contents (0.75 wt.% for Sicomin resin and 0.5 wt.% for Ebalta resin) simultaneously maximize the static bending strength and interlaminar shear strength. If the benefits obtained at the static level have already been explained above, in terms of ILSS, the literature reports that they are explained by the better interface/interphase bond between the epoxy matrix and carbon fibers [62,63].

In this context, the incorporation of CNFs into the epoxy matrix improves its strength and the interface, increasing the stress transfer and, consequently, the ILSS of the composites. On the other hand, CNF contents higher than the optimal values promote lower interlaminar shear strength due to weak interactions between nanofillers and epoxy resin, as well as the imminent crack propagation from the agglomerates that act as stress concentration points. As mentioned in the literature [47,64], these agglomerations/aggregations reduce the interfacial area between the matrix/CNFs and, consequently, the mechanical engagement of the polymer chains in the nanoparticles. In addition, only a few polymer molecules penetrate between the CNFs due to the increased viscosity [43,51,52].

The effect of strain rate on interlaminar shear strength was also analyzed and the results obtained for the different laminates are shown in Figure 13. This study involved only control laminates (with neat resin) and laminates with nano-enhanced resins whose CNF content maximized the properties studied (0.75 wt.% for the Sicomin resin and 0.5 wt.% for the Ebalta resin). In Figure 13, symbols represent the mean values and the scatter bands represent the maximum and minimum values obtained in the experimental tests. It is possible to observe that higher loading rates promote higher interlaminar shear strength values. In terms of laminates produced with neat Sicomin resin, for example, this increase is around 39.9% (from 43.9 MPa to 61.3 MPa), while for laminates nano-enhanced with 0.75 wt.% of CNFs it is about 44.6% (from 47.3 MPa to 68.4 MPa). These values for laminates produced with the Ebalta resin are 57.1% and 55.4%, respectively. This effect is well-explained in the literature [65,66,67], where lower loading rates provide enough time for the micro-cracks to propagate along the matrix, while at higher rates the micro-cracks have to break the covalent bond established between the matrix/CNFs or even break the nano-fibers for their propagation and, consequently, higher ILSS values arise. Simultaneously, other energy absorption mechanisms can also occur at high loading rates such as shear delaminations, matrix cracking and translaminar fracture [65,66].

Regardless of the temporal effect on the previously mentioned damage mechanisms, for Agirregomezkorta et al. [65] the loading rate effect is essentially due to the viscoelastic nature of the polymeric matrices. Therefore, based on this dependence, and similar to what was established for the bending tests, a linear model can be fitted to the data according to the following equation:(9)$$ILSS=a\;ln\left(x\right)+b$$
where *x* is the displacement rate value, and the constants *a* and *b* are shown in Table 4.

From this table it is possible to note that the proposed model presents good accuracy (Stdev higher than 0.997) and can be used as a forecasting tool to estimate the interlaminar shear strength as a function of different loading rates. Regardless of the resin, it can also be seen that, in terms of ILSS, laminates with nano-enhanced resin by CNFs show more sensitivity to the loading rate than the control laminates (with neat resins). On the other hand, when comparing resins, laminates produced with Sicomin resin have less sensitivity due to greater physicochemical compatibility with the carbon reinforcements. Exposure to hostile environments also affects the mechanical performance due to interaction with composite constituents, essentially at the level of the fiber/matrix interface [68,69,70]. In this context, the interlaminar shear strength is the ideal parameter to evaluate this effect, which is shown in Figure 14 for different solutions (alkaline and acid) and concentrations. From the results obtained, both solutions (HCl and NaOH) affect the interlaminar shear strength of all laminates, and this trend increases with increasing solution concentration.

For example, compared to non-immersed laminates, ILSS decreases around 3.9% for laminates produced with neat Sicomin resin and immersed into HCl at 35 wt.% and by about 7.4% for laminates immersed into NaOH for the same concentration. However, when this resin is reinforced with CNFs these values are 10.7% and 14.7%, respectively. The same comparison for laminates produced with Ebalta resin leads, respectively, to values of 11.2% and 13.6% for laminates with neat resin and 10.7% and 11.8% for nano-enhanced laminates. Based on this analysis, it can be noted that the laminates produced with neat Sicomin resin are the least sensitive when exposed to both solutions, but when the resins are reinforced with CNFs the ILSS values approach each other significantly and mitigate the difference found between laminates with neat resins. In fact, according to Agirregomezkorta et al. [65], the interlaminar shear strength depends essentially on the mechanical performance of the resin, but it is also mentioned in the literature that ILSS decreases with exposure to hostile environments [43,71]. Moreover, this effect increases with the pH value [70,72]. Finally, another point of evidence expressed by the ILSS results obtained for both resins is related to the greater severity of the alkaline solution in relation to the acidic one (see Figure 14). This is in line with the literature and, as shown in Figure 15, is justified by the greater severity of the damage introduced in the laminates [43,44].

These damage mechanisms are relative to the laminates with neat Ebalta resin; however, they are representative of those observed for the other laminates studied. Furthermore, they are similar to those observed in Figure 11 but with higher severity. In this context, it is possible to observe several delaminations resulting from the contact between longitudinal and transverse yarns (B), matrix cracking (D) and broken fibers (E). Similar to what was observed above, cracks also occur into resin-rich regions. The observed severity of the damage mechanisms previously reported is a consequence of the absorption, penetration and reaction that occur between solutions and composite constituents [69]. In terms of the fiber/matrix interface,, the degradation is due to dehydration of the matrix and penetration of solutions through micro-cracks [70,73], crazes or voids in the matrix [68]. Simultaneously, the matrices are also attacked under the combined action of water diffusion and the presence of H^+^ (promoting matrix expansion and production of pits/micro-cracks), while the fibers are attacked with consequent cracks on their surface. In this context, the composite’s strength is significantly affected [32,74].

The effect of temperature was also analyzed, and Figure 16 shows the results obtained for samples immersed into distilled water and into both solutions (NaOH and HCl) at concentrations of 15 wt.%. The immersion time was 20 days, and the fluids were at temperatures of 22 °C (room temperature), 40 °C and 80 °C.

It is possible to observe for all laminates and fluids that increasing the temperature decreases the ILSS values. For example, compared to non-immersed laminates, the interlaminar shear strength decreases around 3.9% when laminates produced with neat Sicomin resin are immersed into distilled water at room temperature, and this value increases to 13% when the temperature increases to 80 °C. These values are, respectively, 4.9% and 17.9% for the same laminates reinforced with CNFs. On the other hand, when the Ebalta resin is considered, these values are around 18.4% and 26.9% for laminates with neat resin, and 7.1% and 12.4% for laminates with nano-reinforced resin, respectively. When the same analysis considers HCl, these values are 1.5% and 47.1% for laminates with neat Sicomin resin and 7.3% and 48.3% when nano-reinforced, while for the Ebalta resin they are 5.6% and 46.7% for laminates with neat resin and 3.8% and 45.1% when nano-reinforced, respectively. Finally, the same comparative analysis for the alkaline solution leads to ILSS values slightly lower than those observed for the acidic solution (consequently higher percentage values relative to the non-immersed laminates), evidencing its higher severity mentioned above.

These results are in line with those discussed based on Figure 14, evidencing the fact that laminates with neat Ebalta resin are the most affected by the solutions analyzed, but the opposite trend occurs when they are nano-reinforced (higher benefits are obtained than with Sicomin resin). In this context, the resin used has an influence on interlaminar shear strength, confirming what was reported by Banna et al. [75]. Another clear point of evidence reveals that increasing temperature promotes a further decrease in interlaminar shear strength for all laminates. In fact, temperature accelerates the degradation of mechanical properties, and both acid and alkali solutions are more harmful than water at the same temperature, which agrees with the studies developed by Yang et al. [30]. This is explained by the fact that diffusion is a thermally activated process and, in this context, increasing temperature accelerates short-term diffusion and increases the diffusion coefficient [76]. Consequently, the liquid flow into laminates increases. Furthermore, there are residual stresses at the interface that are responsible for the appearance of micro-voids/micro-cracks due to different coefficients of thermal expansion existing between fiber and matrix [77]. In this context, the aggressive fluids create hydrostatic pressure at the crack tips and hasten crack propagation and damage in the matrix [78]. The acid solution and the alkali solution more strongly damaged CFRPs compared with aqueous water at the same temperature [30].

## 4. Conclusions

Bending tests and interlaminar shear tests were carried out to investigate the behavior of carbon laminates nano-reinforced with CNFs. Relevant improvements in bending stress (20.4 and 12.5% for Sicomin and Ebalta, respectively) and bending stiffness (13.8 and 8.8% for Sicomin and Ebalta, respectively) were obtained with the incorporation of CNFs. A reduction in bending strain of the carbon laminates is observed with the increasing filler content for both epoxy resins, showing a more brittle behavior. In terms of strain-rate sensitivity, the bending stress increases for all laminates with an increasing strain rate. For the laminates with Sicomin resin, relative to the control laminates an increase in bending stress of 22.7% was obtained, while that of the laminate with 0.75 wt.% of CNFs increased by 15.4%. The values obtained for bending modulus were 7.4% and 6.3% for Sicomin resin, and 3.9% and 18.4% for Ebalta resin. Nano-reinforced laminates produced with Sicomin resin have a lower strain-rate sensitivity than laminates produced with Ebalta resin. A linear model is proposed as a tool that describes with high precision the evolution of bending stress, bending stiffness and bending strain for all laminates.

In terms of ILSS, the best results coincided with the results of the static bending point, that is, Sicomin resin in laminate with 0.75 wt.% CNFs and Ebalta resin with 0.5 wt.% CNFs. For all laminates studied, the ILLS is strain-rate sensitive; the values increase for higher values of strain rate. Additionally, a linear model describes accurately its behavior for all laminates. In both resins, the nano-enhanced laminates with the optimal percentage of CNFs show higher results as an answer to strain-rate sensitivity.

The corrosive environments significantly affect the ILSS response, and their effects are strongly dependent on the concentration of the solution. The alkaline solution promotes higher decreases in ILSS than the acid solution. However, Sicomin resin is less sensitive to both solutions than those produced with Ebalta resin. When CNFs were added to the resins, a decrease in ILSS was observed in comparison with laminates produced with neat resin. This effect is more significant in Sicomin because the adhesion between the CNFs and this resin is poor.

Finally, regardless of the epoxy resin or the environment in which the laminates were immersed in solution or water, higher temperatures induced a decrease in ILSS. The addition of CNFs was beneficial as it reduced the degradation of the laminates.

## Figures and Tables

**Figure 1 materials-16-04332-f001:**
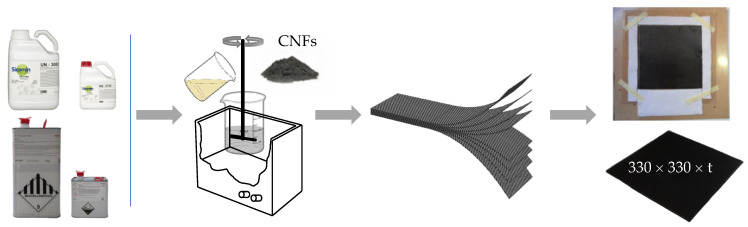
Schematic sequence of the manufacturing process of composite laminates with nano-reinforced resins.

**Figure 2 materials-16-04332-f002:**
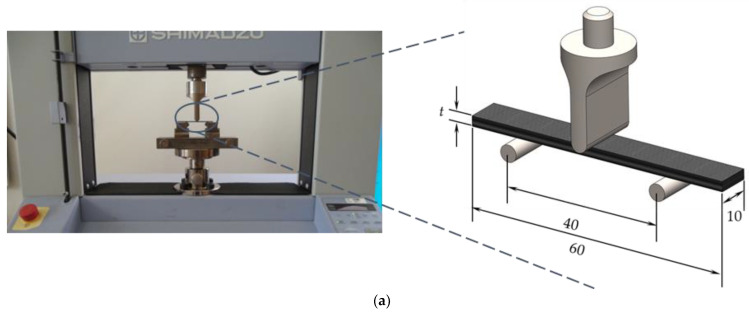
Apparatus, schematic view and geometry of the specimens for (**a**) three-point bending tests; (**b**) interlaminar shear tests. All dimensions in mm.

**Figure 3 materials-16-04332-f003:**
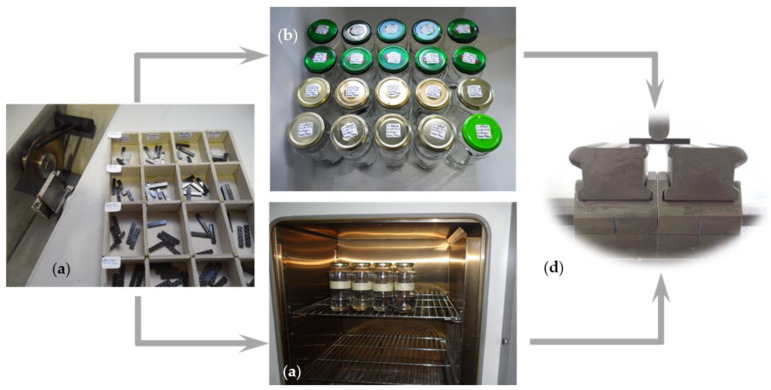
(**a**) Specimens after manufacture; (**b**) specimens immersed into different hostile solutions at room temperature; (**c**) oven with specimens inside to study the temperature effect; (**d**) ILSS tests.

**Figure 4 materials-16-04332-f004:**
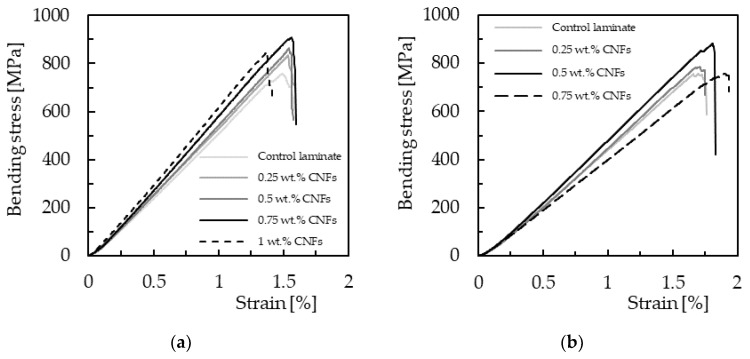
Bending stress–strain curves for laminates with (**a**) Sicomin resin; (**b**) Ebalta resin.

**Figure 5 materials-16-04332-f005:**
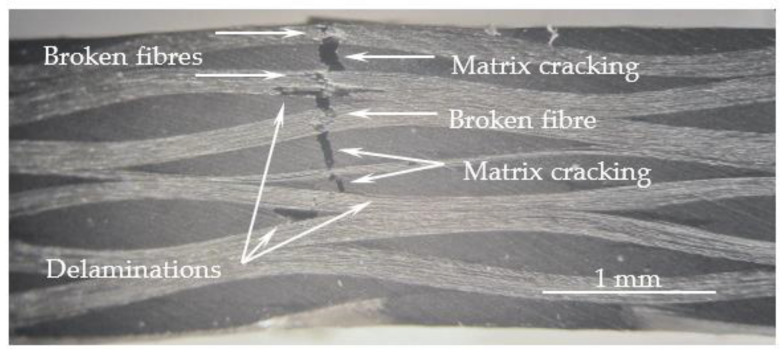
Typical failure mode observed for all composites.

**Figure 6 materials-16-04332-f006:**
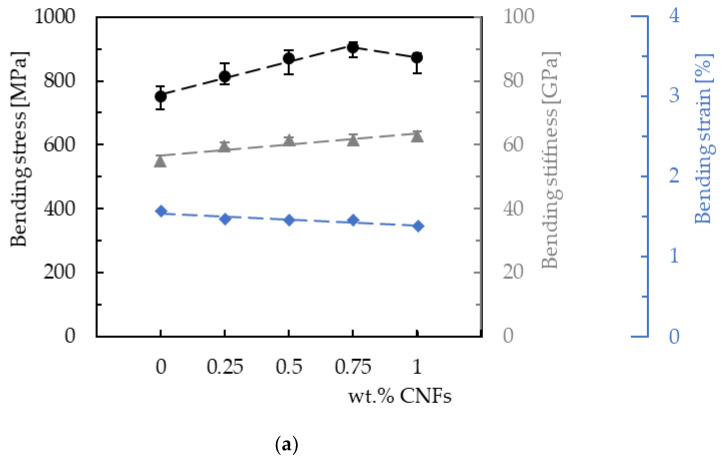
Bending properties for laminates produced with (**a**) Sicomin resin; (**b**) Ebalta resin.

**Figure 7 materials-16-04332-f007:**
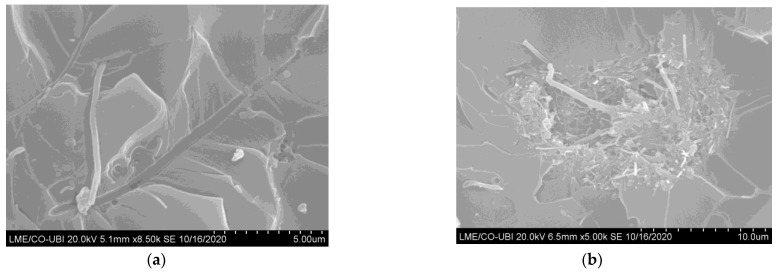
SEM pictures for the Ebalta resin with (**a**) 0.5 wt.% of CNFs; (**b**) 0.75 wt.% of CNFs [39].

**Figure 8 materials-16-04332-f008:**
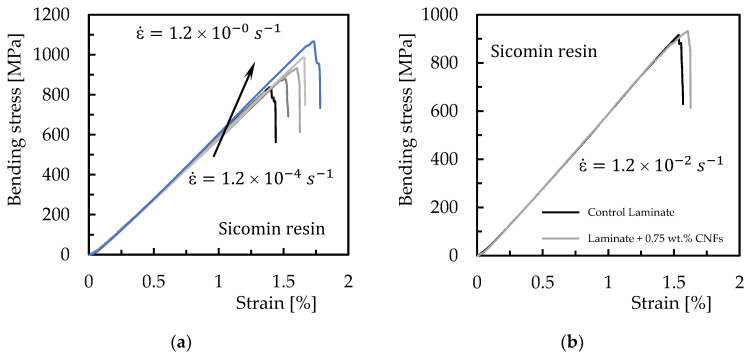
Bending stress versus strain curves for (**a**) laminates produced with Sicomin resin reinforced with 0.75 wt.% of CNFs and different strain rates; (**b**) comparative curves obtained at 1.2 × 10^−2^ s^−1^ for control laminates and laminates produced with Sicomin resin reinforced with 0.75 wt.% of CNFs.

**Figure 9 materials-16-04332-f009:**
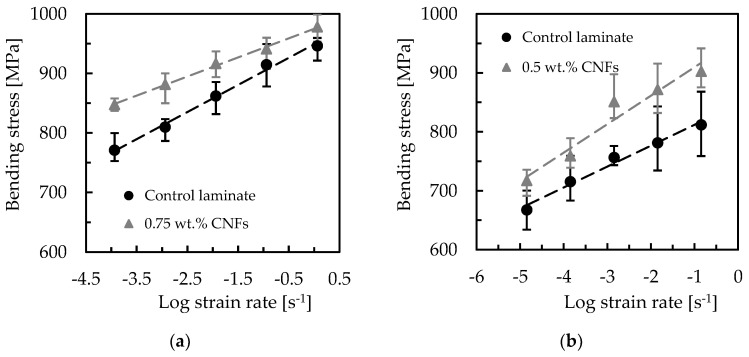
Strain rate effect on (**a**) bending stress for Sicomin resin; (**b**) bending stress for Ebalta resin; (**c**) bending stiffness for Sicomin resin; (**d**) bending stiffness for Ebalta resin; (**e**) bending strain for Sicomin resin; (**f**) bending strain for Ebalta resin.

**Figure 10 materials-16-04332-f010:**
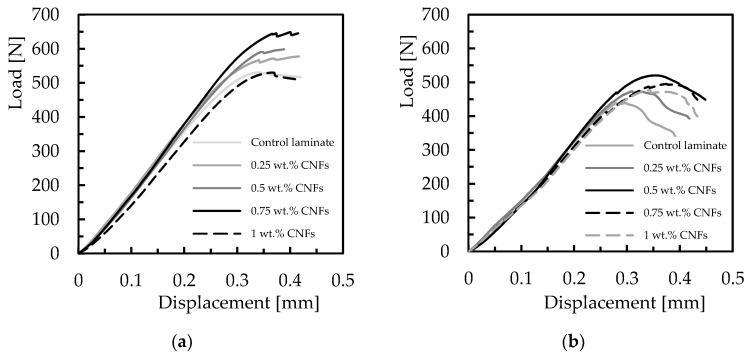
Typical load versus displacement curves for laminates with (**a**) Sicomin resin; (**b**) Ebalta resin.

**Figure 11 materials-16-04332-f011:**
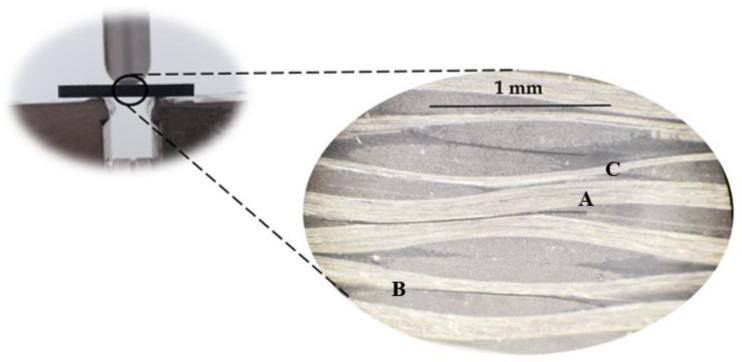
Damage mechanisms observed for laminates with neat Ebalta resin.

**Figure 12 materials-16-04332-f012:**
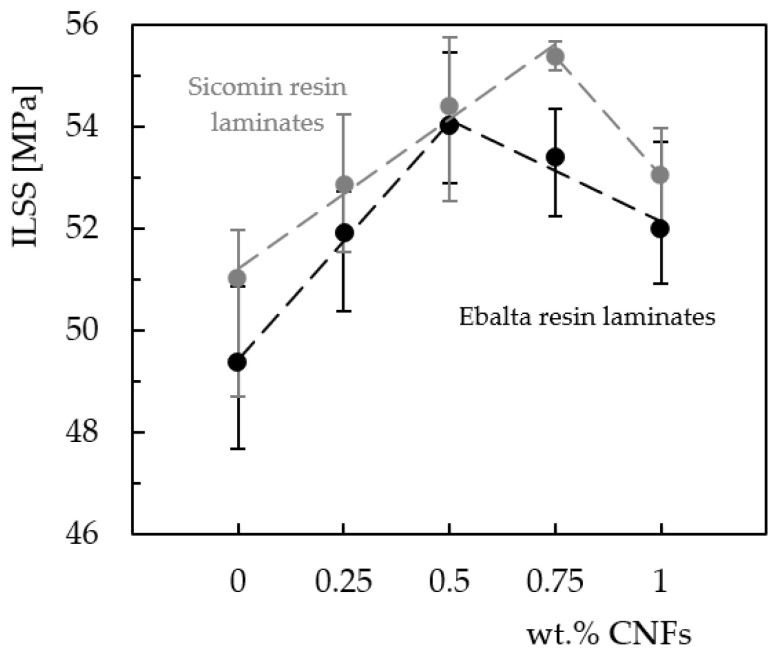
Effect of CNF content on interlaminar shear strength, for two epoxy resins.

**Figure 13 materials-16-04332-f013:**
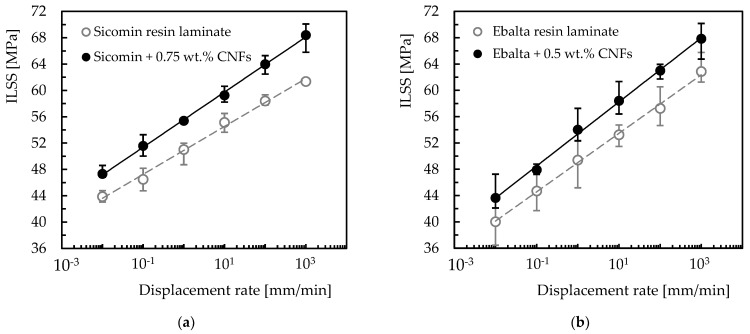
ILSS values at different strain rates for laminates with: (**a**) Sicomin resin; (**b**) Ebalta resin.

**Figure 14 materials-16-04332-f014:**
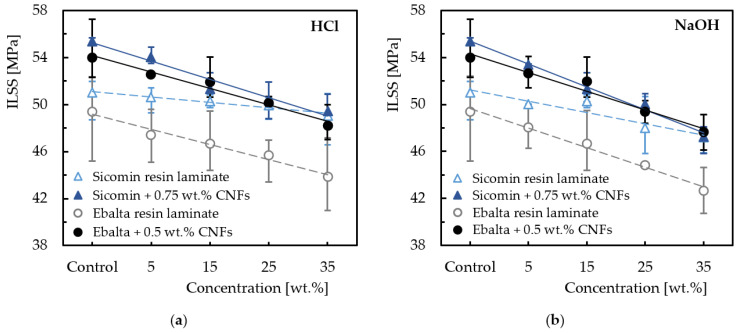
Effect of the hostile solution and its concentration on the ILSS for (**a**) HCl; (**b**) NaOH.

**Figure 15 materials-16-04332-f015:**
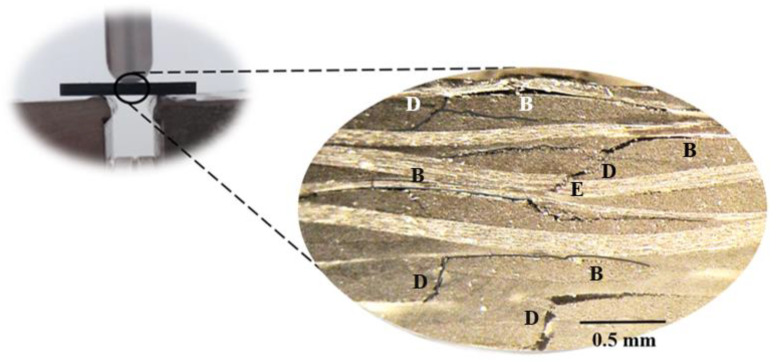
Damage mechanisms observed for laminates with neat Ebalta resin immersed into NaOH at 35 wt.%.

**Figure 16 materials-16-04332-f016:**
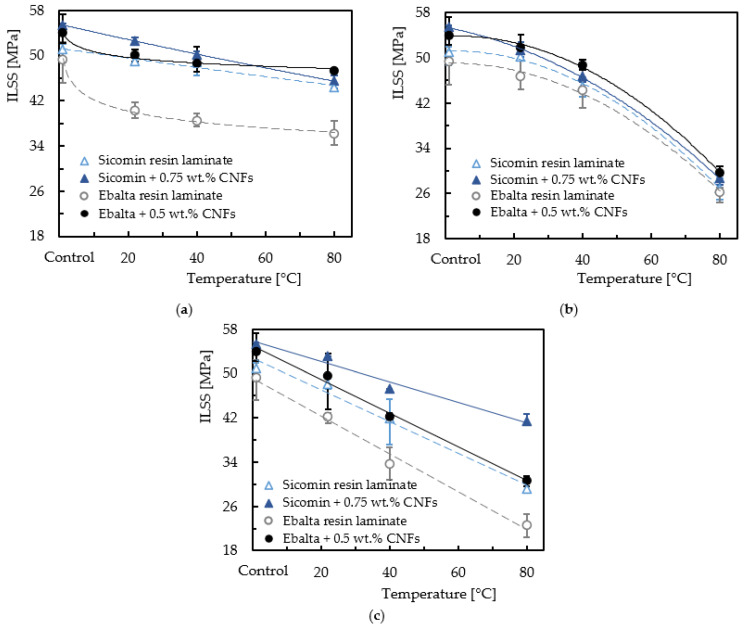
Temperature effect on the interlaminar shear strength for laminates immersed into (**a**) distilled water; (**b**) hydrochloric acid (HCl); (**c**) sodium hydroxide (NaOH).

**Table 1 materials-16-04332-t001:** Main mechanical and physical properties of the epoxy resins.

Property		Sicomin SR 8100/SD 8824	Ebalta AH 150/IP430
Colour		Light yellow liquid	Opaque
Viscosity (at 25 °C)	[mPa×s]	285 ± 60	250 ± 50
Density at 20 °C	[g/cm^3^]	-	1.13 ± 0.02
Modulus of elasticity	[N/mm^2^]	2970 ± 280	3400 ± 300
Maximum strength	[N/mm^2^]	108 ± 1.1	125 ± 1.2
Elongation at max. load	[%]	4.9 ± 0.2	-
Elongation at break	[%]	11.8 ± 0.3	5.9 ± 0.1
Charpy impact strength	[kJ/m^2^]	52 ± 4	60 ± 6
Glass transition/DCC	[°C]	63 ± 3	-
Water absorption 48 h/70 °C	[%]	1.2 ± 0.2	-

**Table 2 materials-16-04332-t002:** Summary of all tests performed and conditions analyzed in this study.

Properties/Conditions	Sicomin Resin Reinforced (wt.%)					Ebalta Resin Reinforced (wt.%)				
	0	0.25	0.5	0.75	1	0	0.25	0.5	0.75	1
Static tests										
Bending properties	X	X	X	X	X	X	X	X	X	X
Strain rate effect	X	(^a^)				X	(^a^)			
Interlaminar shear tests										
ILSS	X	X	X	X	X	X	X	X	X	X
Strain rate effect	X	(^b^)				X	(^b^)			
HCl—concentration effect	X	(^b^)				X	(^b^)			
NaOH—concentration effect	X	(^b^)				X	(^b^)			
HCl—temperature effect	X	(^b^)				X	(^b^)			
NaOH—temperature effect	X	(^b^)				X	(^b^)			
Distilled water—temperature effect	X	(^b^)				X	(^b^)			

(^a^)—Only for the CNF content that maximizes the bending properties. (^b^)—Only for the CNF content that maximizes the ILSS.

**Table 3 materials-16-04332-t003:** Parameters of the equations that fit the effect of the strain rate.

Laminate	Properties	Parameters		Stdev
		*a*	*b*	
Sicomin control laminate	Bending stress (*σ*)	948.94	45.52	0.997
	Bending modulus (*E*)	63.65	1.12	0.998
	Strain bending (*ε*)	1.70	0.108	0.997
Sicomin laminate + 0.75 wt.% CNFs	Bending stress (*σ*)	974.94	32.05	0.998
	Bending modulus (*E*)	64.77	0.967	0.995
	Strain bending (*ε*)	1.63	0.089	0.992
Ebalta control laminate	Bending stress (*σ*)	847.30	35.44	0.983
	Bending modulus (*E*)	48.75	0.424	0.951
	Strain bending (*ε*)	1.91	0.066	0.970
Ebalta laminate + 0.5 wt.% CNFs	Bending stress (*σ*)	957.77	48.29	0.972
	Bending modulus (*E*)	57.70	2.23	0.993
	Strain bending (*ε*)	1.94	0.039	0.994

Stdev = standard deviation.

**Table 4 materials-16-04332-t004:** Parameters of the equation describing the loading rate effect on the ILSS.

Laminate	ILSS Equation Parameters		Stdev
	*a*	*b*	
Sicomin control laminate	1.58	50.89	0.997
Sicomin laminate + 0.75 wt.% CNFs	1.82	55.55	0.999
Ebalta control laminate	1.93	49.01	0.999
Ebalta laminate + 0.5 wt.% CNFs	2.12	53.36	0.999

Stdev = standard deviation.

## Data Availability

Not applicable.

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
