# Peer review of "Effect of Carbon Nanofibers on the Strain Rate and Interlaminar Shear Strength of Carbon/Epoxy Composites"

_materials, 2023, doi:10.3390/ma16124332_

Round 1
Reviewer 1 Report
Comments to Authors
1. Highlights some key points.
2. Write keywords in alphabetical order.
3. Please mention the Graphical abstract.
4. Introduction, revise it and could be elaborated in terms of what other mechanism that has been used in previous or other related studies?
5. Add the impact of current work on industry and future research.
6. carbon nanofibers (CNFs). Mentioned the same format throughout the manuscript.
7. Figure 9. Mentioned the original dimension.
8. For °C, Figure, Table, % mentioned same format throughout the manuscript.
9. Authors need to improve the problem statement in the introduction section.
10. Why and how the said parameters were selected for this work? More specific details needed to be added with the use of the latest reference.
11. In your discussion section, please link your empirical results with a broader and deeper literature review.
12. Do not cite more than three references at the same time.
13. Use Endnote or Mendeley for reference. The current format is not according to the prestigious journal.
14. The conclusion is very lengthy, make it short.
15. Authors need to increase the literature and problem statement from the current recent papers such as.
v https://doi.org/10.1007/s10924-022-02561-8.
v https://doi.org/10.1016/j.envres.2023.115253.
The authors should focus on grammatical errors throughout the manuscript.
Author Response
Detailed answers to the reviewer
The authors are grateful for the work of the reviewer and the opportunity to improve the article. New text entered the manuscript is highlighted in blue coluor.
“1. Highlights some key points.”
Answer: Thank you. The highlights are:
- Laminates with epoxy resin reinforced with CNFs shows significant improvements in terms of bending stress and bending stiffness.
- Alkaline solution promotes higher ILSS reductions and the addition of CNFs is not beneficial.
- Immersion in water and high temperatures promote a decrease in ILSS but the addition of CNFs reduces the degradation.
“2. Write keywords in alphabetical order.”
Answer: Thank you. As suggested the correction was made.
“3. Please mention the Graphical abstract.”
Answer: Thank you. As suggested the Graphical abstract was added.
“4. Introduction, revise it and could be elaborated in terms of what other mechanism that has been used in previous or other related studies?”
Answer: Recent bibliography has been added and the text was improved (Lines 158-174).
Several immersion aging studies of CFRP composites in water, acidic and alkaline solutions at different temperatures show that degradation adversely affects the mechanical properties and, according to thermal and mechanical analysis, ageing depends on the ageing temperature and the ageing medium, being more pronounced at higher temperatures, mainly in acidic conditions [29,30]. Uthaman et al. [29], attributed the decreases in properties of the composites to the degradation of the resin matrix and debonding at the fib-resin interface. On the other hand Yang et al. [30], showed that the addition of multi-walled carbon nanotubes (MWCNTs) improves the ageing resistance of CFRP due to good interfacial interaction and their high barrier property. Nanoclays, in turn, improve the ageing resistance of CFRP due to their high aspect ratio and moderate interfacial adhesion. In short, CFRP containing nanofillers reduces the loss of mechanical properties less than pure CFRP. Kojnoková et al. [31] studied how different chemical environments at a given temperature affect the viscoelastic properties of composites when subjected to degradation by immersion and confirmed a synergistic influence caused by degradation changes and a plasticising effect due to water absorption, which causes a reduction in the modulus of elasticity.
- Uthaman, A.; Xian, G.; Thomas, S.; Wang, Y.; Zheng, Q.; Liu, X. Durability of an Epoxy Resin and Its Carbon Fiber- Reinforced Polymer Composite upon Immersion in Water, Acidic, and Alkaline Solutions. Polymers 2020, 12, 614, doi:10.3390/polym12030614.
- Yang, T.; Lu, S.; Song, D.; Zhu, X.; Almira, I.; Liu, J.; Zhu, Y. Effect of Nanofiller on the Mechanical Properties of Carbon Fiber/Epoxy Composites under Different Aging Conditions. Materials 2021, 14, 7810, doi:10.3390/ma14247810.
- Kojnoková, T.; Nový, F.; Markovičová, L. The Study of Chemical and Thermal Influences of the Environment on the Degradation of Mechanical Properties of Carbon Composite with Epoxy Resin. Polymers 2022, 14, 3245, doi:10.3390/polym14163245.
“5. Add the impact of current work on industry and future research.”
Answer: The text was revised and improved (Lines 201-213).
Nowadays, both industry and researchers are faced with new challenges/opportunities on a daily basis, due to the current climate change, the paradigm shift in production/distribution/energy consumption, changes in the way we travel, the need for more protection, both individual and collective, to prevent natural or provoked attacks, the application/development/optimisation of composite materials is topical and requires continuous advancement to respond to all these demands. In the future, advanced materials will be responsible for achieving climate neutrality, thereby boosting the economy through green technologies, the development of sustainable transport and safety in all areas of human endeavour. In this sense, optimised nanocomposite materials will be a very important part of the answer to problems in industrial applications, providing materials that are easier to manufacture in complex shapes, lighter, structurally stronger, corrosion resistant regardless of the environment, low thermal conductivity and have a longer life cycle, reducing their environmental footprint and recycling issues.
“6. carbon nanofibers (CNFs). Mentioned the same format throughout the manuscript.”
Answer: As suggested by the reviewer, authors used the same format throughout the manuscript.
“7. Figure 9. Mentioned the original dimension.”
Answer: As suggested by the reviewer, authors improved this figure.
“8. For °C, Figure, Table, % mentioned same format throughout the manuscript.”
Answer: As suggested by the reviewer, authors used the same format throughout the manuscript.
“9. Authors need to improve the problem statement in the introduction section.”
Answer: A detailed review was made. The text has been improved (Lines 222-231).
This study aims to contribute to improving the scientific knowledge of the effect of adding CNFs as a low cost reinforcement through the application of simple dispersion techniques in composites. It responds to a gap in the literature that does not report studies on the effect of corrosive environments on ILSS strength. Although some studies have investigated the aging behavior of CFRP with nanofillers in different solutions, the study of common aging factors in the industry, such as hydrochloric acid (HCl), sodium hydroxide (NaOH), water and temperature are not frequent. The degradation of the mechanical properties of CFRPs under these different aging conditions was systematically studied in order to understand their behavior and mathematical models are proposed to estimate the properties for complementary conditions.
“10. Why and how the said parameters were selected for this work? More specific details needed to be added with the use of the latest reference.”
Answer: Thank you for the pertinent question. The text has been revised. Lines 325 -330.
In industry there are processes that involve, for example, in the preparation of surfaces for adhesives, and also exposure to aggressive environments. It appears that in the literature it is not frequent to find studies that analyze the behavior and durability of laminates, in particular, with CNF nano reinforcement. Thus, it responds to a gap in the literature that does not report studies on the effect of corrosive environments on ILSS strength. Moreover, in bending testing, the specimen is subjected to a complex combination of forces including tension, compression and shear as it bends. For this reason, flexural testing is easily performed in engineering to evaluate the response of composite materials to realistic loading situations. On the other hand, the ILSS tests allow a measurement of the resistance of the composite to delamination under shear forces parallel to the layers of the laminate and thus to the adhesive/adhesive interface.
“11. In your discussion section, please link your empirical results with a broader and deeper literature review.”
Answer: As suggested by the reviewer, authors improved the discussion section.
“12. Do not cite more than three references at the same time.”
Answer: As suggested by the reviewer, authors did not cite more than three references at the same time.
“13. Use Endnote or Mendeley for reference. The current format is not according to the prestigious journal.”
Answer: Thank you. Mendeley had already been used.
“14. The conclusion is very lengthy, make it short.”
Answer: The conclusions section was revised to focus on the important findings as well as suggested.
“15. Authors need to increase the literature and problem statement from the current recent papers such as.
https://doi.org/10.1007/s10924-022-02561-8.
https://doi.org/10.1016/j.envres.2023.115253.”
Answer: The authors analyzed the articles proposed by the reviewer. The studied materials, nano reinforcements, experimental tests, ageing conditions and analytical models were not approached or discussed in these two works.
“The authors should focus on grammatical errors throughout the manuscript.”
Answer: The manuscript was revised and the reviewer's concern was taken into account.

Reviewer 2 Report
The paper entitled 'Effect of carbon nanofibers on the strain rate and interlaminar shear strength of carbon/epoxy composites' deals with characterization and of static and dynamic mechanical properties. The paper is well-structured and provides significant contribution in the field. Only minor changes are required according to reviewer's opinion.
- Impact tests could also provide additional insight in the mechanical analysis.
The language is good and no extensive editing is required.
Author Response
Detailed answers to the reviewer
The authors are grateful for the work of the reviewer and the opportunity to improve the article. New text entered the manuscript is highlighted in blue coluor.
“The paper entitled 'Effect of carbon nanofibers on the strain rate and interlaminar shear strength of carbon/epoxy composites' deals with characterization and of static and dynamic mechanical properties. The paper is well-structured and provides significant contribution in the field. Only minor changes are required according to reviewer's opinion.
- Impact tests could also provide additional insight in the mechanical analysis.
The language is good and no extensive editing is required.”
Answer: The authors would like to thank you for your time in reviewing their work and for all your opportune suggestion. We can add that a study is currently being prepared to address the issue of low-velocity impact and multi-impact in nano-enhanced laminates with CNFs. Following your suggestion, it would also be interesting to study the laminates in this article subjected to different types of impact. Nevertheless, the manuscript was detailed revised and the reviewer's concern was taken into account.

Author Response
The authors are grateful for the work of the reviewer and the opportunity to improve the article. New text entered the manuscript is highlighted in blue coluor.
Answers to the issues and improvements identified by the reviewer in the attached document.

Reviewer 4 Report
The authors have studied the effects of the composition of carbon nanofibers on the strain rate and interlaminar shear strength of carbon/epoxy composites. This is an interesting work however there are a few concerns in the manuscript which need to be addressed. After careful addressing of the concerns and the submission of the revised version, the manuscript may be accepted for publication in the Materials.
1. The findings given in the abstract and conclusion are expectable facts. You can give quantitative results that may be comparable with other composites.
2. The novelty of the work with respect to the state of the art and the expected/obtained results is not very clear. Authors must write a separate paragraph at the end of the introduction section. Moreover, they can add a paragraph related to the use of fillers in line with this work.
3. The author includes recent references (2020-2022) for better understanding for beginners.
4. What is the contribution of choosing NaOH and HCl?
5. What is the reason for selecting CNF as (0.75 and 0.5 wt.% ) within a limited range?
6. The author marks the mode of failure in Figure 3.
7. The author mentions the characteristics of Ebalta resin and Sicomin resin and also explains the difference between both resins.
8. The author explains while adding above 1 wt.% of CNF, the properties will affect the composites.
9. Significant improvements are needed in the conclusion section. A conclusion must answer your problem statements as well as be supported by the data gathered, followed by a concise take-home message.
10. Detail explanation is needed for the results and discussion section.
11. The author clearly explains effect of CNF in the ILSS.
This is an interesting work however there are a few concerns in the manuscript which need to be addressed. After careful addressing of the concerns and the submission of the revised version, the manuscript may be accepted for publication in the Materials.
Author Response

(The authors gave the same response as above.)

Round 2
Reviewer 3 Report
Accept in present form
Reviewer 4 Report
The authors carried out all the corrections given by the reviewers.
Accepted in Present Form